# From Paper to E-Prescribing of Multidose Drug Dispensing: A Qualitative Study of Workflow in a Community Care Setting

**DOI:** 10.3390/pharmacy9010041

**Published:** 2021-02-16

**Authors:** Anette Vik Josendal, Trine Strand Bergmo

**Affiliations:** 1Norwegian Centre for E-Health Research, University Hospital of North Norway, 9038 Tromsø, Norway; Trine.Bergmo@ehealthresearch.no; 2Section for Pharmaceutics and Social Pharmacy, Department of Pharmacy, University of Oslo, 0316 Oslo, Norway; 3Department of Pharmacy, Faculty of Health Sciences, UiT. The Artic University of Norway, 9037 Tromsø, Norway

**Keywords:** e-prescribing, multidose drug dispensing, community care, interviews, nurses, pharmacists, work, workflow, collaboration

## Abstract

E-prescribing is now widespread and, in some countries, has completely replaced paper prescriptions. In Norway, almost all prescribing is electronic, except for multidose drug dispensing (MDD), which is still sent to the pharmacy by fax or ordinary mail. MDD is an adherence aid used by one-third of all patients receiving home care services. In this paper, we present results from a qualitative study evaluating the introduction of e-prescribing for MDD in a community health care setting. The focus is on the work and workflow for the pharmacists and nurses involved in the medication-handling process. We used the pragmatic process evaluation framework and the systematic text condensation method to analyse the data. We conducted 12 interviews with 34 nurses and pharmacists. This study shows that the e-prescribing of MDD led to greater integration between systems, both within the existing MDD system and across care levels, potentially improving patient safety. However, the structured prescriptions increased the need for clarifications, resulting in an increased overall workload. A greater understanding of the roles and responsibilities of the different professionals in the medication management chain and their needs would improve the workflow of the nurses and pharmacists involved.

## 1. Introduction

Electronic prescribing (e-prescribing) can improve physicians’ workflow, increase pharmacy efficiency, and improve patient safety [1,2,3,4,5,6]. In recent years, the use of e-prescribing has increased significantly in Europe, especially in the Nordic countries, Estonia, and the Netherlands [7]. It is also widely used in the United States, Australia, and Canada [8]. E-prescribing improves the eligibility and clarity of prescriptions [4,5,6], reduces prescribing errors [4], improves coordination, and ensures the privacy and security of personal health information [9]. However, it can also create new errors, such as incorrect dosage instructions, missing information, incorrect product (medicine or strength), and wrong quantity or duration of therapy [1,8,10,11].

In Norway, e-prescribing was implemented in primary care in 2013, and today, over 90% of prescriptions are sent electronically (27 million per year) [12]. The prescriptions are sent directly from the prescribers to a central database called the Prescription Mediator, which stores the prescription until it is dispensed, a physician deletes it, or the prescription expires (usually one year from the date of prescribing). All pharmacies have access to the Prescription Mediator, and the patient can collect the prescriptions from any pharmacy in the country. The only prescriptions still on paper are for individuals receiving prepacked multidose dispensed drugs (MDD).

MDD is an adherence aid used by one-third of patients receiving home care services in Norway [13]. Here, medicines are machine-dispensed in unit-of-use disposable bags, one unit for each dose occasion [13,14]. The MDD system is used in the Nordic countries and the Netherlands [15]. Compared to patients with ordinary prescribing, patients with MDD have fewer serious drug–drug interactions in their medication lists [16,17] and higher medication adherence [18,19] but are also more prone to medication errors in care-level transitions and inappropriate prescribing [17,20,21,22].

In the current paper-based system, the general practitioner (GP) prints out a list of the patient’s medication treatment and faxes it to the pharmacy. This printed list is the MDD prescription and is valid for one year [23]. The pharmacy staff manually transfer the medicines information from the paper prescription into the electronic MDD system at the pharmacy. They then order MDD from the manufacturer, who packs the medicines and sends them to the home care services, along with a paper copy of the prescription list. The nurses manually enter the medicines information from this list into their own electronic record system [23]. In addition to being very time-consuming, all these manual steps in the medicine management process increase the risk of medication errors [8]. Concerns have also been voiced about duplicate prescriptions when GPs prescribe an electronic prescription in addition to the paper prescription list [24].

There is now an ongoing effort in Norway to implement e-prescribing for MDD. Here, the paper medication list is replaced by individual e-prescriptions for each medication, which are stored in the existing Prescription Mediator [25]. From the implementation of e-prescriptions, we know that technical standards, system design, and training are important foundations to realise the full potential of e-prescribing [1] but also that these systems can create new workarounds because of inconsistent use, computer systems, and network limitations [1,9]. No studies have analysed the transition from paper to e-prescribing of MDD. However, the e-prescribing of MDD has the potential to improve the work and workflow for the staff involved. Going from paper and fax to electronic transmission of prescriptions should reduce manual entry work, which, in turn, should reduce prescription errors [8,26]. Moreover, e-prescribing can minimise interruptions from phone and fax communications, improving work efficiency [27,28,29].

The first pilot test of e-prescribing of MDD began in 2016, the second in 2018. At the time of writing the current paper, 26 GP offices, 2 pharmacies, and 4 home care districts in the southern part of Norway use the system. The current study is part of a larger case study exploring how the e-prescribing of MDD affects patient safety and the health professionals’ work, experienced benefits, risks, and challenges. We have previously investigated how e-prescribing affects the GPs involved; the participants found the new system to be less time-consuming than paper and fax, hence improving workflow and efficiency [23]. This current study investigates how the e-prescribing system affects the work and workload for the home care nurses and pharmacists involved.

## 2. Materials and Methods

### 2.1. Research Design

We used a qualitative research design to explore how the work and workload of nurses and pharmacists in a primary care setting are affected by e-prescribing of MDD. We used a pragmatic process evaluation approach [30] taking advantage of the real-world setting to identify core experiences from the implementation process. The focus was on how the health professionals experienced the change during the early phase of disseminating the e-prescribing system.

We used a theme-centred interview guide with open-ended questions. The interviews were semi-structured, and the order of topics and questions varied, depending on the informants’ responses. How the e-prescribing affected their work before, during, and after the start-up, experienced benefits, risks, and challenges and how they perceived the impact on patient safety were topics for the interviews. We emphasised that both positive and negative experiences were of interest. In the current paper, we focused on how the new system affected work and workload for the nurses and pharmacists involved in the MDD process (Table 1).

### 2.2. Recruitment and Setting

The national health authorities are responsible for implementing the e-prescribing system, but the process has been slower than expected due to technical difficulties. We took a pragmatic approach to recruitment, aiming to recruit all professionals involved in the implementation. The recruitment started in 2016 and is still ongoing. In this study, we focused on the pharmacists and the home care nurses. We consecutively sent e-mail invitations to designated contact persons at each site, asking them to forward the invitations to relevant health personnel. The invitations briefly described the project and main themes of the interviews. One reminder was sent to the non-responders. We sent invitations for follow-up interviews 10 months to 2 years after start-up.

A total of 26 nurses and 8 pharmacists accepted our invitation to be interviewed (see Table 2). We conducted 12 in person interviews; 7 group interviews with up to 6 participants at a time, and 3 individual interviews with one pharmacist working in homecare and two nurses/nurse managers. The choice of individual or group interviews was based on practical and time-saving reasons for those interviewed. We considered that groups would provide sufficient depth to the information we wanted to collect, and in addition, the informants could stimulate each other to provide experiences and views. Differences in views and experiences were as interesting as uniform opinions among the informants. We also sent invitations to participate in follow-up phone interviews. Only one community pharmacist in the home care service and two pharmacists at one of the pharmacies accepted and were interviewed a second time.

### 2.3. Data Collection

Two researchers completed the group interviews, which took place at the participants’ workplace. Phone interviews were conducted for practical reasons, and were completed by one researcher. The interviews lasted 30–45 min and were recorded on tape and transcribed by a professional agency. We stored audio recordings separate from the anonymised transcribed data material.

### 2.4. Data Analysis

Both authors (a pharmacist and a registered nurse) read, discussed, and structured the transcribed material and participated in the analysis of the data. The systematic text condensation (STC) method described by Malterud [31] was used to conduct a thematic analysis of the meaning and content of the data across cases. STC is based on principles of Giorgi’s psychological phenomenological analysis [32] and includes a descriptive approach presenting the experience as expressed by the participants themselves [31]. The procedure consisted of the following steps:Reading of the material several times to obtain an overall impression (from chaos to themes);The identifying and sorting meaning units representing different aspects of the research question, and perform coding and sub-coding for these (from themes to codes);Condensation and summarising the content in the coded groups (from code to meaning); andDeveloping descriptions reflecting the participants’ important experiences (from condensation to descriptions and concepts).

The transcripts were coded to maintain the content using NVivo 12 software. We created nodes that were arranged in a coding tree with three recurring themes: local workflow changes, change in collaboration and communication pattern, and increased access to information. To achieve trustworthiness, the researchers engaged in an ongoing process of discussion and reflection throughout the process of analysis. We focused on changes in workflow that persisted after the initial start-up and have not included challenges and problems directly related to the transition. We used the Consolidated Criteria for Reporting Qualitative Research (COREQ) checklist to ensure the structure and style of this manuscript.

### 2.5. Ethics

The Data Protection Officer at the University Hospital of North Norway approved the project (Project No. 02003). All participants voluntarily accepted to be interviewed. The participants received both written and oral information about the study; they were informed about the reason for conducting the research, the researchers’ roles, credentials, and experiences; about anonymity; and that they could withdraw from the study at any time. The data was handled according to local security requirements. To ensure anonymity of the home-care service pharmacist and the nurse manager in the interviews, we only included the setting when presenting quotes.

## 3. Results

The results section presents the findings from the analysing process described above. Changes in the workflow were categorised into three major themes:Local workflow changes;Change in collaboration and communication patterns; andIncreased access to information.

### 3.1. Local Workflow Changes

For the pharmacists, one of the biggest advantages of the new system was the direct transfer of prescriptions into the pharmacy dispensing programme. This eliminated the first step in the paper-based workflow: sorting incoming faxes. Because the e-prescriptions were ordered by date, the pharmacist always knew which prescription was the current one, and the prescription-check could be moved from close to the order deadline to about two days prior to the deadline. This gave the pharmacist more time to do necessary clarifications with the GP. A third advantage with the electronic transfer was that any supplementary documentation that used to be on paper (e.g., applications for compassionate use) was now also electronic.

However, the pharmacists said that correcting prescriptions was more difficult and time-consuming. In one of the interviews, the e-prescribing system was described as limiting the pharmacist’s ability to perform professional judgements and their ability to correct obvious errors made by the GPs:


*“We are more vulnerable if the GP makes a mistake on the e-prescriptions (…) on a regular paper list we can make the change ourselves and ask the GP to sign it afterwards, and when we have received a signature, the problem is solved (…), but now (with e-prescribing) we must have a completely new e-prescription (if the medicines are to be dispensed)” Pharmacy 2*


Specifically, the pharmacists reported having to manually change the item number a lot more frequently. For each active substance, there are only a few item numbers that are dispensable as MDD. In the paper-based system, the pharmacist chose this item number when entering the prescription into the dispensing program. Now, the item number is transferred automatically from the electronic prescriptions. The nurses reported some patients who did not get their regular medications dispensed as MDD because the item number on the e-prescriptions was not dispensable as MDD. This was a particular problem in the first few months. When the pharmacy staff became aware of this problem, they created new routines to check for MDD-dispensable item numbers of the same medicine and manually change the prescriptions.

Because of missing medicines, some home care nurses had started to double-check the MDD deliveries. They reported that the extra checking led to staffing problems; they went from one to four nurses on delivery days after e-prescribing was introduced. Some emphasised that all the double-checking increased their responsibility for the patients’ drug treatment. They further said that these new routines increased patient safety compared with the old system:


*“Before (e-dose), it was not necessary to check if, for example, the prescriptions needed a renewal – we did not have to check if there were any medications missing on the prescription list. Before it was it was the norm that all medicines were included, but now we have to check all the time that ok, the medicine is missing. Why is it missing?” Home service 4*


However, some nurses also experienced that e-prescribing led to faster changes of the MDD bags, which meant fewer manual corrections of the bags. They suspected that this was both because the process of transferring prescriptions was faster and also because the GPs sent the updates during the consultations rather than waiting until the end of the day:


*“It is our experience that if they do it electronically, then it is much easier, because then we know that it arrives quite quickly. With those who do not have e-prescribing, it may be that they have not sent that fax, or the fax has not arrived and then they have to send it by post. So, the (paper) process takes a lot more time” Home service 1*


### 3.2. Change in Collaboration and Communication Patterns

One of the built-in changes with the new system was electronic messaging between the pharmacy staff and the GPs. The pharmacists said that the GPs replied faster and that there were fewer misunderstandings when communication was electronic compared to fax. In addition, they said the messages could be sent and replied to at more convenient times so that they did not disrupt the GP in the middle of a patient consultation:


*“We get answers (from the GPs) quickly. We do not have to fax, call and all that, it saves quite a lot of time. In addition, we get answers to what we actually asked about to a much greater extent.” Pharmacy 1*


Despite these improvements, the pharmacists agreed that communicating with the GPs took more time overall. This was because they did more clarifications, especially regarding the renewal of prescriptions. The former paper prescription list was valid for a one-year supply of medicines, needing only an annual renewal. With the new system, each medicine had an individual e-prescription. Although one pharmacist expressed relief over not having to renew the prescription list by fax anymore, all the informants agreed that individual e-prescriptions were more time-consuming. First, it led to more inquiries about renewals; second, when a prescription was renewed, it needed to be checked by a pharmacist. One pharmacist explained that this was especially problematic because the GPs frequently prescribed very small quantities, even though the patients used them regularly:


*“…we have to contact the GP much more frequently and say: you have prescribed medications for ten days, a multi-dose roll is for 14 days, and we need more tablets.” Pharmacy 2*


Prescription renewals were also one of the main workload concerns of the nurses. Whenever a prescription was updated, both the GPs and pharmacy staff sent messages to the home care nurses. These messages had to be checked to see if there was an actual change in the medication treatment. Because the number of messages had increased, so had the nurses’ workload. The nurses also considered information about renewals that were irrelevant and voiced concerns that the increased number of messages without useful information made them overlook the relevant information:


*“We get a lot of information that we do not really need, which only creates a lot of work (for us). I think this can cause dangerous situations in relation to whether we are able to administer the medicines that we are obliged to.” Home service 4*


Both the pharmacists and many of the home care nurses also said that the renewal of prescriptions had led to many unnecessary phone calls between the two because the home care nurses did not have access to the Prescription Mediator, where the e-prescriptions were stored. One nurse said that:


*“…we call the pharmacy and ask what prescriptions are there, and we also have to call the GPs a lot to check what they have included…. Before, we had it (the prescription list) physically in our hands, and there was no doubt which prescriptions were valid.” Home service 2*


### 3.3. Increased Access to Information

Storing all e-prescriptions in the same database had several implications for our informants. The pharmacists were notified of all changes in the patient’s medicines treatment, even when the prescriptions were not dispensed in MDD. Though this resulted in them doing more prescription checks, they also noted that this change seemed to improve patient safety:


*“The disadvantage is that (...) all transactions in the Prescription Mediator are flagged as a change for us. On paper multidose, this course of antibiotics would never have been noticed because it was not reported to us. So, the safety is much better with electronic multidose because you will notice all changes. But the workload increases. It increases a lot.” Pharmacy 1*


Storing MDD prescriptions and ordinary prescriptions in the same place also implied that the pharmacists had direct access to prescriptions from doctors other than the GP. This was particularly beneficial after hospital visits. In the paper-based system, the e-prescription could not be used to dispense MDD, and the pharmacists would wait for the GP to fax an updated prescription list. In the new system, they had a valid prescription to dispense right away. However, when the GPs updated the prescription list after a hospital visit, this could also cause new problems because the hospital doctors and GPs had different prescribing privileges. The GP and hospital doctor could also prescribe the same medication to the same patient or the GP could renew prescriptions without withdrawing the old ones. This led to outdated and duplicate prescriptions in the Prescription Mediator, and the pharmacists used more time to identify which prescription was the correct one. In one of the interviews, the pharmacists described one such situation:


*“P1: They (the GP) had not renewed the prescriptions, they had just prescribed new ones, and the old ones remained. P2: Three prescriptions on one medication and three on another, you know. P1. Maybe both from a specialist and from the GP, and another one from the GP...(…) P3: Yes, for many patients the Prescription Mediator is full of clutter.” Pharmacy 1*


A last implication of using a common database for e-prescriptions was that regular community pharmacists now had access to the MDD prescriptions. This increased the pharmacist’s workload because the patients or their caretaker could collect the prescriptions. When it was time to order MDD, there were no valid prescriptions, and they had to contact the home care service and the GP to get new ones. In addition, both the home care nurses and the pharmacists expressed concerns that patients might misuse the system and collect addictive medications at their local pharmacy in addition to getting them in MDD.

However, in one of the pharmacist interviews, it was also stated that this might actually improve patient safety. With the paper-based system, the GPs sometimes unintentionally prescribed a medication both as an e-prescription and as a paper-based MDD prescription, which could lead to the patient taking a double dose. With the electronic system, this was no longer possible:


*“The good thing about the addictive medications is that we definitely catch it (now), we cannot dispense multidose if there is no valid prescription (...) We can see that it had been taken out, so in that sense it gives us better control, but it is very inconvenient for us.” Pharmacy 2*


## 4. Discussion

The current study shows that e-prescribing of MDD affects the work of pharmacists and home care nurses. First, it changed the workflow locally at both the pharmacy and home care services. Second, the system changed the collaboration patterns between the different personnel involved. Third, it increased access to the medicines information for health personnel who were not directly involved. Despite many of the steps in the medication management being more time-efficient, the frequency of which they were performed increased, and the overall impression from both the nurses and pharmacists was an increased workload. However, most agreed that the additional work also increased patient safety.

Most of the experiences from the informants—both positive and negative—seem to stem from the fact that the prescribing of MDD and ordinary prescriptions is now closely integrated. More integrated systems mean that the prescriptions and communication pathways are more standardised, information is transferred more quickly between the actors, and access to medicines information is changed. This is similar to the GPs’ experiences, who also described the system as having an in-built safety mechanism, but could be more time consuming because they needed to incorporate prescriptions from other physicians [23,33]. Though the design and interface of the computer systems have been shown to influence workflow [3], and was an important issue for the GPs [33], this was not brought up by the informants in our study.

The most prominent change in terms of workload for both the pharmacists and nurses was the increased frequency of prescription renewals. This is an effect of going from one complete prescription list with one expiry date to individual standardised e-prescriptions of separate dates and quantities. Previous studies have shown that prescribing the wrong quantity of a medicine is among the most common errors in e-prescribing [10,11,34]. However, the GPs involved in the electronic prescribing of MDD also received insufficient training before the transition [23] and this challenge might thus improve with time as the GPs realise the importance of prescribing the correct amounts in the new system.

E-prescriptions have been shown to require more frequent contact with GPs than paper-based prescriptions [35,36]. This might be a result of an increased number of errors but could also be because the standardised prescriptions leave less room for interpretation, thus making corrections more difficult [6,37]. This means that the pharmacist might have to contact the GP to correct errors they would normally correct themselves. In the interviews, both the pharmacists and nurses expressed that they were now more dependent on the GPs and more vulnerable to the GPs’ mistakes.

The fact that the pharmacists find correcting errors more time-consuming might also be because their local workflow had changed. In the paper-based MDD system, the pharmacist corrected many errors while entering the prescription into the dispensing programme. However, because this transfer step was now eliminated, correcting prescriptions became a procedure outside the normal workflow, thus feeling more time-consuming. In fact, it would seem that the standard workflow was improved within the pharmacy in the new system. The pharmacists worked continuously with the changes as they arrived rather than waiting until the deadline, resulting in a more even workload throughout the day. In addition, the standardised prescriptions, together with the automatic transfer, eliminated many of the manual steps in the prescription handling at the pharmacy. When the pharmacists expressed an increased workload in the new system, this seems to be because of a higher number of deviations from the standard workflow and possibly more time-consuming corrections.

For the nurses, faster transfer of the prescriptions and changed access to prescription information seemed to affect the local workflow the most. Because the prescriptions were transferred faster, the nurses found that the medicines were dispensed in MDD quicker, which resulted in fewer manual corrections of the MDD bags. However, the changed collaboration and communication pattern meant that prescriptions were now stored in the Prescription Mediator, which they did not have access to. This increased the need to contact the GP or pharmacy to ask about the patients’ medication use. This need should be reduced when the nurses get access to updated medication lists, a process that is now underway as part of a national e-health strategy [38].

Some home service units also added an extra control step to check the content of the MDD bags, and this increased their workload further. They went from one to four nurses on MDD delivery days. Though this error was corrected after a few months, the nurses continued to double check the bags. This also increased the feeling of being more responsible for the patients’ drug treatment. E-prescribing has previously been linked to increased workload and responsibilities for the nurses involved [39]. This can negatively affect the patients if the nurses are reallocated from other areas involved in direct patient care. Home care services have expanded in recent years but not the staffing levels, and this can affect nurses’ job satisfaction, create adverse advent incidents, and lower patient satisfaction [40,41].

The pharmacists also described an increased responsibility and patient safety. They performed an extended prescription check compared to the paper-based system, and identified potential misuse of narcotics, which would not have been possible previously. Additionally, the system prevented the GP from prescribing an e-prescription and an MDD prescription for the same medication, which had been a challenge with the paper-based system. Common to all these improvements is that they stem from having prescriptions for MDD users and non-MDD users in the same database.

Using the national database also means that the health personnel not directly involved in handling the MDD got access to the MDD prescriptions. This is particularly useful for clinicians, where a lack of access to an up-to-date medication list can cause medication errors [42]. Such errors are especially common during care transitions and occur more frequently for MDD users [21,22]. That hospital doctors can make changes in the MDD bags directly can also reduce discrepancies and errors in the medication lists during care-level transitions.

However, this could also cause misunderstandings and result in potentially dangerous situations for the patients, i.e., patients taking double doses if the old prescriptions are not withdrawn when new ones are issued. Because all physicians could make changes to the MDD prescript, the GPs involved in the e-prescribing system also expressed uncertainty about the status and validity of the current medication list [33]. Ultimately, when more actors can influence prescribing and dispensing, the system becomes more complex. In turn, this increases the uncertainty about what the other actors could and should do, resulting in an increased need for communication between the actors involved.

Communication and collaboration between health personnel is, however, essential for safe medication management, but establishing new routines and collaboration patterns takes time. Because most of our interviews were held only a few months after start-up, these new routines would not yet be fully incorporated. It would thus be difficult to see if there are any long-term benefits of closer collaboration, and this might explain why our informants’ views on increased contact were mostly negative. Future research should focus on collaboration and how this affects the quality of prescribing, drug use pattern, and other aspects of patient safety.

### Strengths and Limitations

This is an exploratory study with a pragmatic approach to recruiting participants. We invited everyone involved in testing the electronic MDD system and included those who accepted our invitation. We interviewed 34 nurses and pharmacists across several sites, and we believe that we have captured the most important experiences of those who have started using the system. However, only two pharmacies from the same pharmacy chain participated. We know that the MDD work practices vary between pharmacies [13] and the experiences of future pharmacists who start using this system might thus differ from what we have found here. We plan to continue to collect data and include more pharmacies as they begin using e-prescribing for the MDD users.

## 5. Conclusions

Our study shows that the e-prescribing system for MDD led to a greater integration between systems, both within the existing MDD system and across care levels. The increased integration has several benefits in terms of faster information exchange and easier access to more complete information about the patients, increasing patient safety. However, the structured prescriptions also meant that it was more difficult for the nurses and pharmacists to correct mistakes. This, in turn, resulted in more frequent contact between the actors involved and increased their workload. In addition, increased integration implies that those not a part of the current MDD system, such as hospital doctors and community pharmacists, could influence the prescribing and dispensing of MDD. A greater understanding of the roles and responsibilities of the other actors in the medication management chain and their needs could improve many of the challenges that the nurses and pharmacists in the present study experienced.

## Figures and Tables

**Table 1 pharmacy-09-00041-t001:** Excerpt from the interview guide relevant for this study.

Aspects of E-Prescribing of MDD Affecting Work and Workload
How does e-prescribing change the way you work? How does e-prescribing change the communication and collaboration with the GPs, home care nurses, or pharmacists? How do you think the new system affects patient safety?

**Table 2 pharmacy-09-00041-t002:** Interview details.

No	Participants	Number of Participants (n)	Setting	Second Interview
1	Pharmacist	3	Pharmacy 1	
2	Nurse	6	Home service 1	
3	Nurse	6	Home service 2	
4	Nurse	5	Home service 3	
5	Pharmacist	1	Home service 4	Yes n = 1
6	Nurse	2	Home service 4	
7	Pharmacist	4	Pharmacy 2	Yes n = 2
8	Nurse	1	Home service 5	
9	Nurse	5	Home service 6	
10	Nurse manager	1	Home service 6	

## Data Availability

The data presented in this study are available upon reasonable request from the corresponding author. The data are not publicly available due to ethical reasons.

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
