# Peer review of "From Paper to E-Prescribing of Multidose Drug Dispensing: A Qualitative Study of Workflow in a Community Care Setting"

_pharmacy, 2021, doi:10.3390/pharmacy9010041_

Round 1

Reviewer 1 Report

Thank you for the submission.  It is quite interesting and well-written, with the limitations appropriately acknowledged. The e-prescribing system clearly has both positives and negatives. The aspect of perhaps most concern and warranting some more analysis/discussion is the observation that “Some home service units also added an extra control step to check the content of the MDD bags, and this increased their workload further. They went from one nurse before the implementation to four nurses on MDD delivery days.” This is a major labour impost on a stretched workforce. It was mentioned that this was due to missing doses. I think the authors need to address this concern in a little more detail e.g. why didn’t the pharmacists prevent these errors? What was the root cause? Did the GPs have current medication lists for these patients? Secondly, the discussion should have a brief comparison of the views of GPs (collected previously) and the nurses/pharmacists on the e-prescribing system for MDD. Were there any significant differences in perceptions?

Author Response

Thank you for these thoughtful comments and constructive suggestions. 

The errors the nurses experienced was mostly related to errors during the transition, but the control-step seemed to persist also after the implementation. The problems related to the transition is outside the scope of this study and we have not gone into detail regarding the cause of these errors. We have written more clearly in the methods section that this has not been our focus, and the discussion, at page 10, now reads:

“Some home service units also added an extra control step to check the content of the MDD bags, and this increased their workload further. They went from one to four nurses on MDD delivery days. Though this error was corrected after a few months, the nurses continued to double check the bags. This also increased the feeling of being more responsible for the patients’ drug treatment.”

We have now discussed the GPs views in the first paragraph in the discussion, and have also compared their experiences with that of the nurses/pharmacists throughout the discussion where relevant topics have been discussed.

We appreciate the time and effort that you have dedicated to providing constructive suggestions which help to improve the quality of this manuscript.

Reviewer 2 Report

Here these are my comments:

Abstract

Please add the details about the setting in which data was collected and also mention what qualitative research approach has been used for data analysis.

Introduction

This seems fine, but there is a lack of information on similar studies in other contexts inside or outside of Norway. By the way, it is expected that the authors use the findings of previous studies in the discussion section to compare their findings.

Methods

This section must be organised and the related details should be presented under the following subheadings and with full practical details to help repeating the research process: Design (which research design for data collection and analysis has been selected and why), Sample and setting (the details of sampling and recruitment), Data collection (how data collection has been performed in practice), Data analysis (what data analysis approach has been used and what steps has been taken to reach the data analysis products), Rigour (all considerations of trustworhiness in your research), Ethics (all measures to ensure the ethical research practice). 

Do not forget please that your description above should be accompanied with relevant citations to support the research process.

With group interview, do you mean focus group? please specify and mention all details for group dynamics in the data collection process.

The interview guide should also be presented.

The Malterud is not a quite recognised data analysis (thematic analysis) method in the international literature. If you insist citing it, please provide the full details and steps for data analysis for it.

Lines 117-120, seems to be your findings, but I do not know why it hass been presented before 'results'. They should be transfered.

Results

At the end of each quotation, there should be: gender and age and speciality of the person; for instance (male, 45 years old, pharmacist).

Lines 160-163, please bring a direct quotation to support this statement.

Lines 191-197, again a  direct quotation to support this statement is needed.

Lines 198-201, please describe if they were happy with making frequent phone contacts.

Lines 219-241, a lot to say, but no direct quotations. What is the sourcse of these statement?

Discussion

This section should be improved as you need to present your findings theme by theme and discuss theme by the findings of other studies in other contexts inside and outside Norway.

Lines 261-264, please check the sentence structure. Also, the informants have not discussed on training and education, but you as the interviewer could ask about it to explore this important section of findings.

Lines 267-299, and lines 317-327, many sentences without discussing findings with literature.

The main point missing in your discussion is ''interdisciplinary collaboration for medicines management''. Why everybody in this system believes that her/his own job is negatively affected through making more communication with the other healthcare provider involved in medicines management? Or for instance, the workflow is hindered if more connections should be made? At least you writing and findings give such a meaning to readers. Please read articles published in 'Pharmacy journal' to find how much such a collaboration and close contact has been emphasised to impeove the safety of medicines management, and some articles have been published by Norwegian researchers. This should be attended in your discussion that medicines management is a collaborative task and frequent connections can make the medication practice safer.

Line 359, "...depended more on each other and were more vulnerable to mistakes." This is a problematic sentence and against the common notion that independence in medicines management causes the system safer, but collaboration and frequent contact can endange safety.

You should find the appropriate Equator (https://www.equator-network.org/reporting-guidelines-study-design/qualitative-research/) and try to improve the presentation style of your article using the checklist. Fill out the checklist and attach it to your article, when you resubmit. 

Author Response

Reviewer’s comment: Abstract

Please add the details about the setting in which data was collected and also mention what qualitative research approach has been used for data analysis.

Reply: We have included details on the qualitative research approach in the abstract.

Reviewer’s comment: Introduction

This seems fine, but there is a lack of information on similar studies in other contexts inside or outside of Norway. By the way, it is expected that the authors use the findings of previous studies in the discussion section to compare their findings.

Reply: We have included previous studies in both the Introduction and the Discussion. We have reviewed the literature and have included studies of e-prescribing and studies describing the effect and experiences with the MDD system. We have more than 40 references on the topics. However, there exists no other studies that have analysed the transition from paper to e-prescribing of MDD so our research are novel. We have clarified this on Page 2.  Please let us know if there are some relevant paper on our topic that we have missed.

Reviewer’s comment: Methods

This section must be organised and the related details should be presented under the following subheadings and with full practical details to help repeating the research process: Design (which research design for data collection and analysis has been selected and why), Sample and setting (the details of sampling and recruitment), Data collection (how data collection has been performed in practice), Data analysis (what data analysis approach has been used and what steps has been taken to reach the data analysis products), Rigour (all considerations of trustworhiness in your research), Ethics (all measures to ensure the ethical research practice).  Do not forget please that your description above should be accompanied with relevant citations to support the research process.

Reply: We have re-organised the section as suggested with subheadings and appropriate citations. See pages 2-4.

Reviewer’s comment: With group interview, do you mean focus group? please specify and mention all details for group dynamics in the data collection process.

Reply: We have specified details for the group interviews on page 3 subsection 1.2

Reviewer’s comment: The interview guide should also be presented.

Reply: We have included the theme form the interview guide relevant for this study on page 3 Table 1.

Reviewer’s comment: The Malterud is not a quite recognised data analysis (thematic analysis) method in the international literature. If you insist citing it, please provide the full details and steps for data analysis for it.

Reply: Malterud is often cited in the medical and health service research literature.This method is based on principles of Giorgi’s psychological phenomenological analysis and includes a descriptive approach presenting the experience as expressed by the participants themselves. The method is described in more detail on page 4, section 1.4

Reviewer’s comment: Lines 117-120, seems to be your findings, but I do not know why it hass been presented before 'results'. They should be transfered.

Reply: This has been transferred to Result on Page 4

Reviewer’s comment: Results

At the end of each quotation, there should be: gender and age and speciality of the person; for instance (male, 45 years old, pharmacist).

Reply: We have included ‘setting’ (i.e. pharmacy 1) when presenting quotes. Due to data protection requirements we not specify any more personal details of the persons interviewed. This is to ensure anonymity of the home-care service pharmacist and the nurse manager in the interviews. We have also added an explanation of this in the Ethics section. Se Ethics on page 4 and the Result section.

Reviewer’s comment: Lines 160-163, please bring a direct quotation to support this statement.

Reply: We have included a direct quotation to support this statement on page 5.

Reviewer’s comment: Lines 191-197, again a  direct quotation to support this statement is needed.

Reply: We have included a direct quotation to support this statement on page 6.

Reviewer’s comment: Lines 198-201, please describe if they were happy with making frequent phone contacts.

Reply: The respondents were not happy with too many unnecessary pane calls. This has been clarified on Page 6, last paragraph.

Reviewer’s comment: Lines 219-241, a lot to say, but no direct quotations. What is the sourcse of these statement?

Reply: We have included quotations on page 7. 

Reviewer’s comment: Discussion

This section should be improved as you need to present your findings theme by theme and discuss theme by the findings of other studies in other contexts inside and outside Norway.

Reply: We find that these topics are intertwined, which makes it difficult to discuss them one by one. E.g. when the pharmacists described correcting prescriptions to be more time-consuming, this can either be seen as change in local workflow (the e-prescriptions/software makes it more difficult) or as a result of increased number of errors from the GP. To keep the flow in the discussion, and to avoid repeating information, we prefer having the discussion as one continuous text rather than divided into sections according to the themes. However, we have now made sure that we use the exact name of the themes in the discussion, so that it becomes clearer where the different themes are discussed.

Reviewer’s comment: Lines 261-264, please check the sentence structure. Also, the informants have not discussed on training and education, but you as the interviewer could ask about it to explore this important section of findings.

Reply: We have asked about training and education in the interviews, however the focus in this study is on lasting changes in workflow, we consider this to be outside the scope. We have therefore deleted these sentences from the manuscript.

Reviewer’s comment: Lines 267-299, and lines 317-327, many sentences without discussing findings with literature.

Reply: We have restructured and shortened down both these paragraphs. See page 8.

Reviewer’s comment: Why everybody in this system believes that her/his own job is negatively affected through making more communication with the other healthcare provider involved in medicines management? Or for instance, the workflow is hindered if more connections should be made? At least you writing and findings give such a meaning to readers. Please read articles published in 'Pharmacy journal' to find how much such a collaboration and close contact has been emphasised to impeove the safety of medicines management, and some articles have been published by Norwegian researchers. This should be attended in your discussion that medicines management is a collaborative task and frequent connections can make the medication practice safer.

Reply: Though we agree that interdisciplinary collaboration can improve patient safety for patients this was not a topic discussed by our informants. Their impression was that this system increased their workload, mostly due to “unnecessary” phone calls or other communication, which did not add value in terms of patient safety. However, they did appreciate that the system, by design, increased patient safety. It did this by reducing manual steps in the prescription handling and eliminating certain errors that had been troublesome in the paper-based system. To address this important issue about interdisciplinary collaboration we have however added a new paragraph about future research that discusses the importance of good collaboration. Page 10 last paragraph before “strength and weaknesses”

Reviewer’s comment: Line 359, "...depended more on each other and were more vulnerable to mistakes." This is a problematic sentence and against the common notion that independence in medicines management causes the system safer, but collaboration and frequent contact can endange safety.

Reply: We agree and have deleted the sentence.

Reviewer’s comment: You should find the appropriate Equator (https://www.equator-network.org/reporting-guidelines-study-design/qualitative-research/) and try to improve the presentation style of your article using the checklist. Fill out the checklist and attach it to your article, when you resubmit. 

Reply: We have used the checklist to improve our paper. The COREQ checklist is enclosed as an Appendix. 

We are grateful for these comments and suggestions and appreciate the time and effort that you have dedicated to providing constructive suggestions which help to improve the quality of this manuscript.

Thank you

Round 2

Reviewer 2 Report

Nothing more.